# Physiological effects of alfaxalone anesthesia on rhesus monkeys during intravenous glucose tolerance testing

Kelli L. Vaughan[1]*, Kielee Toepfer[2], Julie A. Mattison[1]

1 Translational Gerontology Branch, Intramural Research Program, National Institute on Aging, National Institutes of Health, Baltimore, Maryland, United States of America, 2 Charles River, Frederick, Maryland, United States of America

* kelli.vaughan@nih.gov

**Data Availability Statement:** All data required to replicate the reported findings of this study are openly available in the OSF data repository, DOI 10. 17605/OSF.IO/JGFM.

## Abstract

Laboratory animal research with nonhuman primates (NHPs) requires anesthesia for most procedures to ensure safety and consistency in sample collection. However, anesthesia drugs can have adverse effects on the physiological measures of interest. Alfaxalone, most notably used in dogs and cats, offers rapid onset, short duration of action, and has a high safety margin. Here, we compared our current anesthesia protocol using Telazol, to three different doses of alfaxalone during a one-hour intravenous glucose tolerance test, the standard evaluation of glucose metabolism in NHPs. Results indicate there are no differences in the rate of glucose metabolism, anesthesia depth measurements, or total duration of sedation, but induction, number of supplemental doses required, and recovery time to eating were affected by the different doses of alfaxalone. Cardiovascular measures showed variability between the four protocols in respiratory rate and systolic blood pressure rates only. These results indicate that alfaxalone can produce a reliable state of anesthesia, similar to our current protocol, and confers minimal cardiovascular or metabolic disturbance, as well as enhanced recovery characteristics. As such, alfaxalone is a promising anesthetic for use in laboratory animals and further investigation is warranted.

## Introduction

In laboratory animal science, anesthesia protocols with minimal adverse side effects are essential for accurate and reproducible experimental outcomes. Ketamine, Telazol® CIII (Zoetis, Kalamazoo, MI), benzodiazepines, alpha-2 agonists, or their combinations are examples of sedation methods that are widely used in lab animal research. Though it has a wide safety margin, ketamine can lead to muscle tensing, which increases the difficulty of many procedures that require the patient to remain still and can lead to post-operative muscle soreness or damage. Telazol is a nonnarcotic injectable anesthetic agent that is formulated with equal parts tiletamine, a dissociative agent, and zolazepam, a benzodiazepine with minor tranquilizing properties [1]. Benzodiazepines are central nervous system depressants; introduction can assist in the relaxation of a patient, but strongly affect the central nervous system and can lead to

**Funding:** This research was financially supported entirely by the Intramural Research Program of the National Institute on Aging, NIH. The funders had no role in study design, data collection and analysis, decision to publish, or preparation of the manuscript. KLV and JAM are employees of the NIH. This project was fully funded by the NIA, though the authors received no specific funding for this work.

**Competing interests:** The authors have declared that no competing interests exist.

inhibiting respiratory drive [2]. Alpha-2 agonists can be added to induce relaxation along with the sedative but should be avoided in animals with significant disease because they may cause dose-dependent peripheral vasoconstriction and cardiovascular depression through reduced cardiac output and blood perfusion and may reduce insulin secretion and glucose utilization [3].

Rhesus macaques (*Macaca mulatta*) are the most commonly used nonhuman primate (NHP) in translational research [4]. With aging phenotypes that closely mimic those of humans, the ability to study rhesus monkeys into older age is invaluable for unraveling the complex interaction of age and age-related diseases. Our laboratory uses rhesus macaques to study age-related changes by measuring aging phenotypes, characterizing longevity, and evaluating novel treatment modalities.

Metabolic dysregulation is frequently used as an indication of overall health status in laboratory animals. Additionally, glucose tolerance testing (GTT) is used to assess metabolic outcomes for intervention studies and as a diagnostic tool for type 2 diabetes mellitus. Unfortunately, common anesthetic agents can have a profound impact on glucose regulation by decreasing insulin release or the sensitivity of the receptors. For example, Isoflurane, a safe inhalant anesthetic, directly inhibits insulin secretion from pancreatic β cells and impairs glucose tolerance [5]. We previously tested dexmedetomidine because it has a fast onset and can be reversed quickly making it ideal for clinical events; however, it decreased insulin secretion and glucose utilization so is not a viable option for GTTs [6].

Alfaxalone (3α-Hydroxy-5α-pregnane-11,20-dione) is a synthetic neuroactive steroid with anxiolytic, anticonvulsant, and anesthetic properties and is derived from its positive allosteric modulation of gamma aminobutyric acid type A receptor [7, 8]. In 2012, alfaxalone was reformulated into an aqueous solution, using β-cyclodextrin as the solubilizing agent, and FDA approved and marketed as Alfaxan® (Jurox Pty. Ltd., Rutherford, NSW, Australia). With its rapid onset, short duration of action, and large therapeutic index, alfaxalone is an excellent option for research animals [9–11]. Reported advantages of this drug include reduced cardiovascular depression, rapid and transitory anesthetic effect after intravenous (IV) or intramuscular (IM) administration and proved to be non-irritant and non-cumulative in muscle [7, 12]. The IV route can be used for initial sedation or as an IV drip for continuous sedation [7].

Because of its relative safety and the manufacturer's recommended use in compromised patients, our goal is to evaluate alfaxalone for potential use in our NHP colony. Though it is reportedly safe to use in compromised and high-risk animals [13], there is no data available on its effect with a GTT, the standard test for assessing glucose metabolism. Therefore, we aimed to 1) conduct a comprehensive evaluation of the cardiorespiratory effects in rhesus monkeys, 2) compare the anesthetic properties of alfaxalone to our current anesthesia protocols during a one-hour GTT, 3) determine if the drug influences glucose metabolism, and 4) establish an effective dose for NHPs.

## Materials and methods

### Humane care guidelines

This study was carried out in strict accordance with the recommendations in the Guide for the Care and Use of Laboratory Animals [14]. All procedures were approved by the National Institute on Aging Intramural Research Program's Institutional Animal Care and Use Committee.

### Animals

Data were collected from five male and five female rhesus macaques (*Macaca mulatta*) housed at the National Institutes of Health Animal Center. Animals ranged from 5–20 years of age

**Table 1. Demographics of study subjects by sex.**

| Sex | n | Age (yrs) (M ±SEM) | Weight (kgs) (M ±SEM) |
|---|---|---|---|
| Male | 5 | 11.6 ± 1.06 | 9.9 ± 0.28 |
| Female | 5 | 8.2 ± 1.28 | 7.8 ± 0.25 |
| | 10 | 9.9 ± 1.75 | 8.9 ± 0.51 |

(9.9 ± 1.75) with baseline weights between 6 and 12 kilograms (8.9 ± 0.51) (see Table 1; additional details in S1 Table). All animals were housed in standard primate caging, with visual and auditory contact with other animals in the room, and with controlled temperature and humidity and a 12-hour light/dark cycle. Commercially prepared monkey chow was distributed twice per day along with daily food enrichment, and water was available *ad libitum*. Environmental enrichment strategies included objects for sensory, cognitive, and motor manipulation.

## Anesthesia procedures

Intravenous glucose tolerance tests (IVGTTs) were performed for all animals under each of four anesthesia protocols presented in a counterbalanced Latin squares design and with a minimum of two weeks between each episode. The order of anesthesia protocol was randomized, and doses were administered intramuscularly as follows: Telazol (100mg/mL concentration) at 5mg/kg (Tel) and alfaxalone (10mg/mL concentration) at 7.5mg/kg (Alf7), 12mg/kg (Alf12), and 15mg/kg (Alf15).

At each testing period, animals were fasted overnight and then anesthetized using pre-calculated drug doses, removed from their home cage immediately at anesthesia induction, transported to a procedure room and weighed. As an added precaution during this lengthy sedation, a prophylactic treatment with glycopyrrolate (0.15mg/kg, IM) to reduce salivation was given for each anesthetic condition; therefore, the specific effects of glycopyrrolate were not assessed. Glycopyrrolate is an anticholinergic that inhibits salivary gland and respiratory secretions and is used commonly in our lab for excessive salivation. For long procedures it offers the ability to prevent reflex bradycardia, stabilizing heart rate while the animal is sedated [15]. However, glycopyrrolate has also been reported to result in hyperthermia, tachycardia, and cardiac arrhythmia in some individuals [15]. A patient monitoring system (BM3 Elite, Bionet, Tustin, CA) was used to assess heart rate, non-invasive blood pressure, and oxygen saturation; thermoregulation was maintained with a veterinary warming system (Hot Dog®, Augustine Temperature Management, Eden Prairie, MN), and a patent catheter was placed in the saphenous vein. The catheter was periodically flushed with heparinized saline to maintain its integrity throughout the procedure.

## Glucose & insulin measures

Blood samples (3mLs) were collected from the saphenous vein at two baseline time points and again at minutes 1, 5, 10, 20, 30, 40, 50 and 60 following a dose of 300mg/kg of 50% dextrose administered through the IV catheter. At each time point, glucose values were measured and recorded immediately following blood collection from the IV catheter using an Ascensia® Breeze 2 blood glucose monitoring system (Bayer HealthCare, LLC., Mishawaka, IN). Blood samples were centrifuged for serum separation and aliquoted before being stored at -80°C until subsequent analysis using an Insulin ELISA (Mercodia, Uppsala, Sweden).

**Table 2. Scoring system for measuring depth parameters.**

| Score | Spontaneous Movement | Toe Pinch Reflex | Jaw Tone | Limb Manipulation |
|---|---|---|---|---|
| 1 | Whole body | Strongly pulls away | Increased tone | Animal controlled movement |
| 2 | Limb & hand/foot movement | Pulls away immediately | Normal tone | Withdrawn immediately when handled |
| 3 | Mouth or facial movements | Delayed pulling away | Decreased tone | Weakly withdrawn |
| 4 | Twitching | Flexes or extends digits | Minimal tone | Flexes or extends digits |
| 5 | No movement | No movement | No tone | No movement |

Adapted from Lee, Flynt [16] and Sun, Wright and Pinson [17]

### Induction, duration, & depth of anesthesia

Induction of anesthesia was defined as the time between drug administration to the safe removal of the monkey from its cage. Duration was defined as the time between removal from the cage to the point that the monkey was able to sit upwards unassisted. After sitting unassisted, the animal was returned to its home cage where it was monitored every 15 minutes until it voluntarily took and consumed fruit (recovery time until eating).

Depth of anesthesia was assessed by one technician blinded to the anesthetic condition and using parameters adapted from those described in previous studies [6, 16, 17] (Table 2). Parameters included spontaneous movement, toe pinch reflex, jaw tone, and limb manipulation and were assessed at baseline and minutes 10, 20, 30, 40, 50, and 60. For depth measures, each was scored on a scale of 1 to 5 with 1 indicating the monkey was too awake for the parameter to be safely measured. Spontaneous movement was also recorded if it occurred during or outside of the specified time points. Toe pinch reflex was assessed by applying firm pressure to the digit of a hind limb with a hemostat. Jaw tone was determined by manually assessing the tensing of the jaw. For limb manipulation, the technician lifted the monkey's arm to approximately 90 degrees with one hand and then released it, allowing it to fall onto the technician's other hand.

The need for supplemental dosing was continuously evaluated throughout the IVGTT. If at any point animals began to move or scored 3 across all depth parameters, a supplemental dose of anesthesia drug was administered and recorded. Under the Telazol condition, ketamine (5mg/kg, IM) was used for supplemental dose (or boost) and alfaxalone (2mg/kg, IM) was used as boost for all the alfaxalone conditions. To keep the technician blinded to the experimental anesthetic condition, all supplemental doses were precalculated and drawn up in advance into 3ml syringes, mixed with sterile water if needed, and labeled with a code.

### Physiological parameters

For the duration of each procedure, technicians monitored rectal temperature, oxygen saturation (SpO2), non-invasive systolic and diastolic blood pressure, heart rate, and respiratory rate. Salivation was noted for being present or absent for each of the anesthesia conditions. Values recorded at baseline and minutes 10, 30, and 60 were used for statistical analysis.

### Statistical analysis

Statistical analyses were performed using SPSS 27.0 software package (IBM SPSS, Inc., Chicago, IL) and graphed using GraphPad Prism version 10.1.2 for Windows (GraphPad Software, Boston, MA). Results were considered statistically significant with $p < 0.05$, and all data are reported as mean +/- SEM, unless otherwise stated. Due to the small sample sizes included in this study, disaggregation of the data by sex would reduce statistical power and the ability to

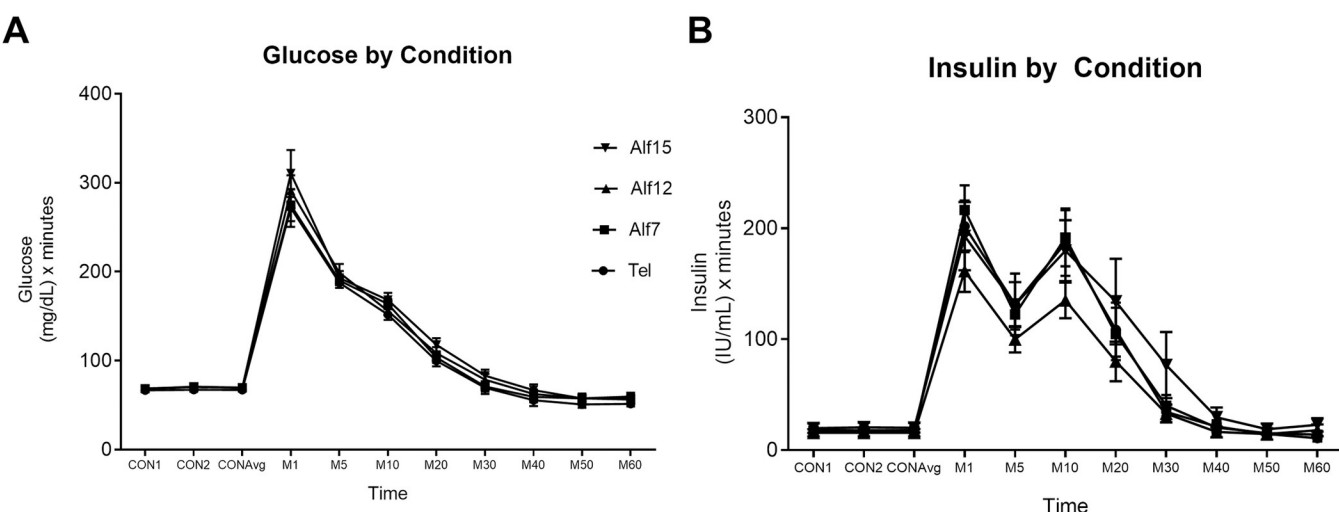

**Fig 1. Changes in glucose and insulin by drug condition during IVGTT.** Neither glucose (A) nor insulin (B) curves differed across drug conditions during one-hour IVGTT. Error bars represent the +/- SEM at each timepoint (n = 10).

detect effects. Therefore, male and female data were pooled for each drug condition. Area under the curve (AUC) was calculated for glucose and insulin values assessed during each one-hour IVGTT and statistical significance was determined based on repeated-measures analysis of variance (ANOVA). Nonparametric Kruskal-Wallis ANOVA were employed to statistically assess depth of anesthesia measurements for each time point across anesthesia conditions. A repeated measures ANOVA was conducted to determine if the time until the first boost of anesthesia was needed during IVGTT testing significantly differed between the four anesthesia conditions and to compare time to eating. Fitted mixed models were used to compare cardiorespiratory parameters for each time point across anesthesia conditions. Finally, the nonparametric Friedman test to compare matched groups was used to compare induction, duration, number of supplemental doses and salivation during each drug condition.

## Results

### Effects on IVGTT outcomes

Comparison of the area under the curves revealed no difference between drug conditions for glucose or insulin during the 60 min IVGTT (Fig 1).

### Sedation characteristics

For induction, the test of differences among repeated measures rendered a Chi-square value of 12.24, which was significant (p = 0.0066), indicating a significant between group difference. Mean time to induction for Tel, Alf7, Alf12, and Alf15 was 4.2, 4.7, 2.9, and 2.1 minutes, respectively. Post hoc analysis with Dunn's multiple comparison test revealed a significant reduction in the time to induction between Alf7 and Alf15 following adjustments for multiple comparisons ($Z = 2.858$, $p = 0.026$) (Fig 2A). Total duration of sedation episode did not differ between drug conditions. Overall differences for elapsed time from anesthesia induction to the first required supplemental dose (time to 1st boost) of ketamine or alfaxalone were statistically significant across anesthesia conditions: $F(3,27) = 4.044$, $p = 0.017$ (Fig 2B). Similarly, the number of supplemental doses needed during a one-hour IVGTT differed significantly between Alf7 and Alf15: Chi-squared value of 12.20, $p = 0.007$ (Fig 2C). A significant difference

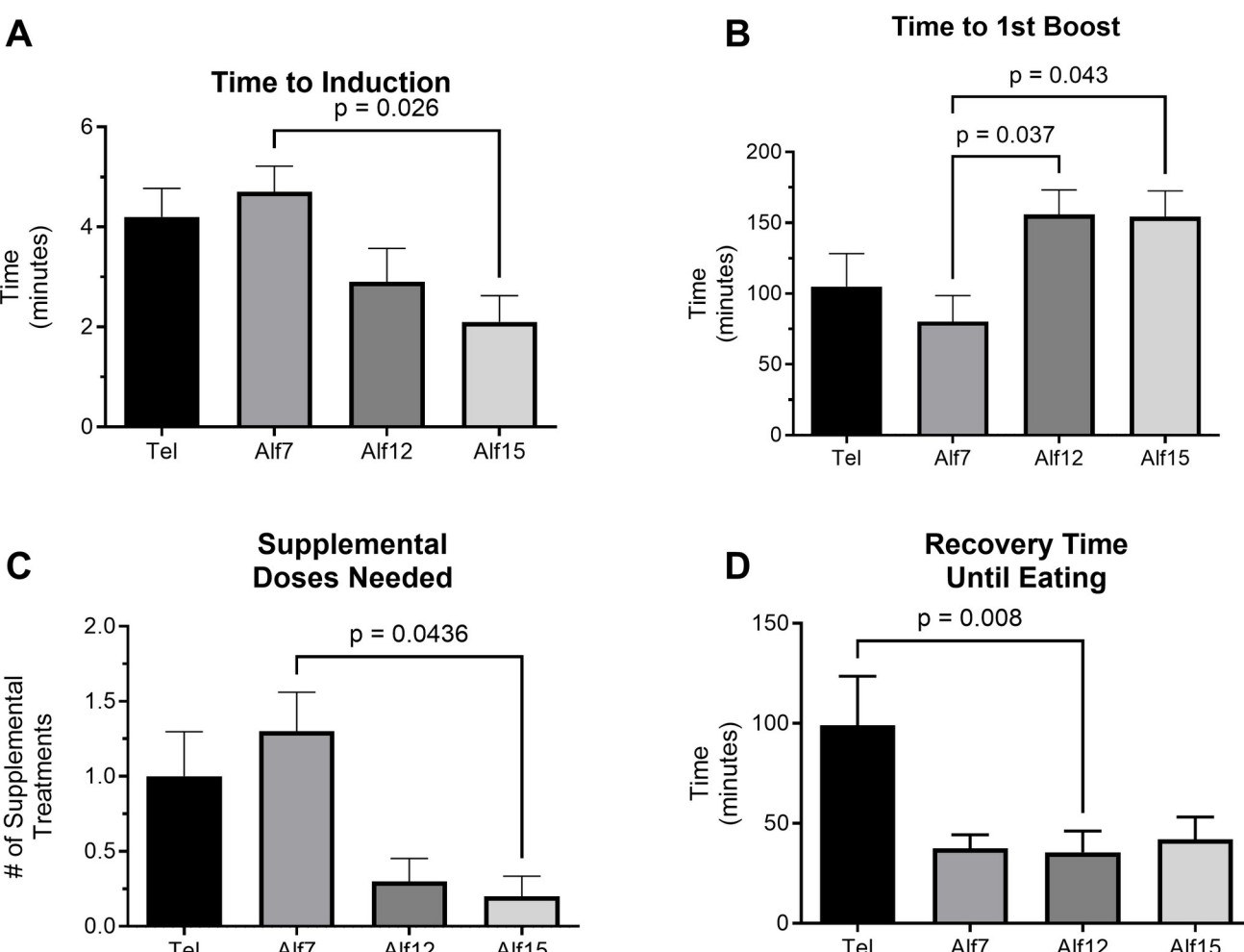

**Fig 2. Characterization of alfaxalone anesthesia in NHPs.** (A) The mean (± SEM) length of time between administration of sedation drug to the time the animal is sedated deeply enough to be safely removed from the cage (n = 10). The mean time to removal from cage was significantly longer for animals sedated with Alf7 compared to animals sedated with Alf15. (B) The mean (± SEM) time until first boost was significantly longer for animals sedated with Alf12 and Alf15 compared to animals sedated with Alf7 (n = 10). (C) The average number of supplemental doses (± SEM) required to maintain a consistent level of anesthesia differed significantly between Alf7 and Alf15 (n = 10). (D) Animals took much longer to eat following recovery if they received Tel compared to Alf12. Time to eat following recovery is displayed as mean (± SEM) (n = 10).

in salivation during sedation between drug conditions was identified ($X^2 = 12.71$, $p = 0.005$), where animals receiving Telazol were more likely to salivate during sedation despite pre-dosing with glycopyrrolate. Likewise, length of time between recovery and eating was statistically significant ($F (1.467, 13.21) = 7.12$, $p = 0.012$) (Fig 2D). Post hoc comparisons reveal a difference between the Telazol condition and Alf12 (adjusted $p = 0.008$).

Regarding depth measures, no between-group differences were found at any of the study time points for spontaneous movement, limb manipulation, jaw tone, or toe pinch.

## Cardiorespiratory parameters

Analyses revealed that heart rate did not differ significantly between drug conditions, though heart rate tended to be lower on Tel compared to the alfaxalone doses ($M_{tel} = 145$ vs. $M_{alf7} = 160$, $M_{alf12} = 154$, and $M_{alf15} = 160$ bpm). There was a statistically significant difference in respiratory rate between at least two drug conditions ($F (3,247) = 4.230$, $p = 0.006$) (Fig 3A).

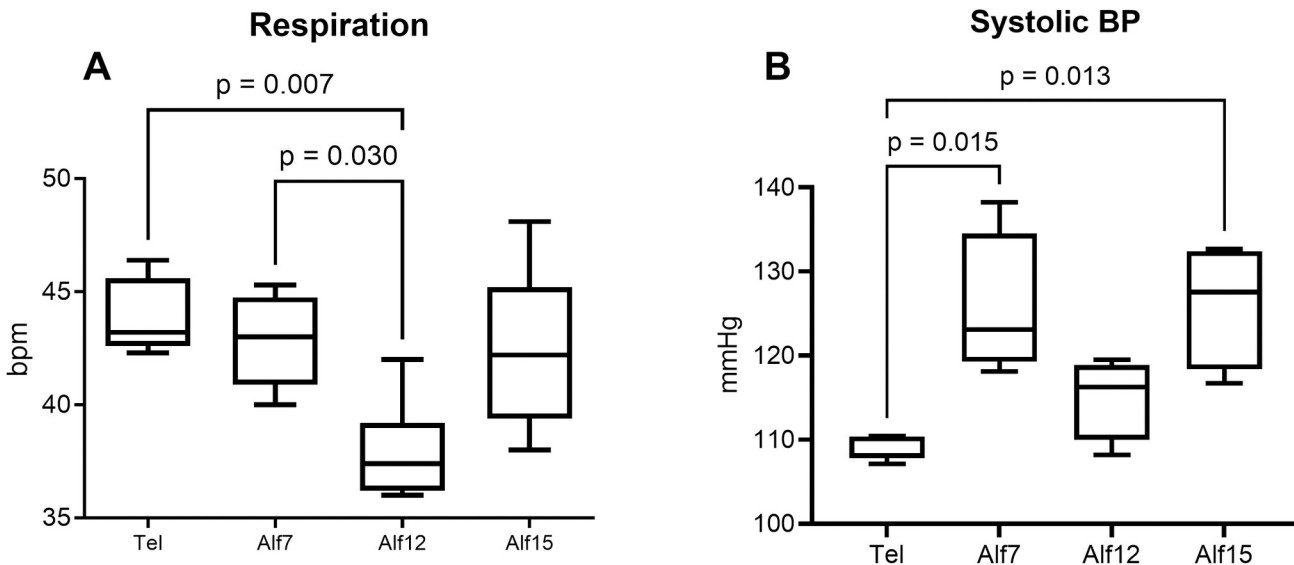

**Fig 3. Cardiorespiratory outcomes of alfaxalone during IVGTT.** (A) The mean (+/-SEM) respiratory rate during IVGTT was significantly lower in the Alf12 condition compared to Tel and Alf7 (n = 10). (B) Systolic blood pressure during IVGTT differed significantly between Tel and Alf7 and Alf15 conditions (n = 10).

Similarly, analyses revealed significant between group differences in systolic blood pressure (SBP) ($F$ (1.875, 16.88) = 4.561, $p$ = 0.028). Here, SBP was lower during Telazol sedation compared to Alf7 and Alf15 (Fig 3B). No significant between groups differences were found in body temperature, SpO2, diastolic blood pressure, pulse pressure, or mean arterial pressure measures.

## Discussion

For the safety of the animals and technical staff, experimental collection of physiological samples from NHPs often requires sedation. Thus, it is essential to ensure that any pharmaceutical intervention does not affect experimental outcomes. To this end, characterizing the effects of all drugs administered for experimental purposes is critical. Although Telazol and ketamine are well-characterized and commonly used in laboratory animal science, they have their own limitations and so, alternative anesthesia drugs are always of interest to researchers.

Unlike our previous trials with dexmedetomidine (4), alfaxalone at any of the three doses showed promise as a suitable anesthetic for use to assess glucose metabolism. This anesthetic agent did not alter the glucose uptake parameters; an important consideration for metabolic testing or use with diabetic animals. Furthermore, alfaxalone was beneficial for reducing the induction time in a dose-dependent manner but did not affect overall anesthesia duration. Here, Alf15 had the fastest induction time, but both Alf12 and Alf15 produced a longer and more relaxed sedation before supplemental treatment was required and with fewer needed anesthesia boosts. These results indicate that at higher doses alfaxalone provides a more consistent and longer lasting sedation compared to Telazol. This finding is similar to Wada, Koyama and Yamashita (18) who report that, in Cynomolgus macaques, higher doses resulted in faster induction and longer duration.

Salivation during anesthesia is a concern and can interfere with technical procedures like intubation or lead to aspiration or other serious respiratory issues. Some animals salivate excessively under anesthesia, challenging the long-term stability needed for an IVGTT, thus it

is our policy to provide glycopyrrolate as a prophylactic. For this study, glycopyrrolate was administered to all animals under each drug condition yet all but one animal salivated while on Telazol. Conversely, only one animal receiving the high dose of alfaxalone salivated during the procedure. This is a promising find for alfaxalone as it may lower the risk potential of aspiration during sedation and may reduce the need for prophylactic treatment for excessive salivation but the specific effect of glycopyrrolate requires additional testing.

Alfaxalone also tended toward a quicker recovery time with Alf12 (35.5 minutes) significantly faster than Tel (99 minutes). There was not a statistical difference in time to first boost and the number of supplemental doses between these two groups, so it suggests an anesthetic regimen effect. However, the overall variability in supplemental dosing precluded analysis, and thus we cannot conclude that the shorter duration was due to the anesthetic regimen or reduced need for supplemental doses of anesthesia. Yet another encouraging alfaxalone outcome as inappetence is common following anesthesia events. For some animals, it can take 24 to 48 hours following a sedation event before feeding behaviors return to normal. This reduction in recovery time is especially important for aged or compromised animals. A quicker return to baseline food intake means less monitoring and supportive care needed following an anesthesia event.

Similarly, depth of anesthesia measures did not differ significantly between drug conditions, suggesting both drugs provide ample anesthesia depth for a one-hour IVGTT. Respiration rate and systolic blood pressure were the only significantly different cardiorespiratory parameters measured. Here, respiratory rate decreased in a dose-dependent manner but began to rebound at the highest dose. Wada, Koyama and Yamashita [18] report similar results but they do not see a rebound, this is likely because the largest dose they tested was 10mg/kg. Our results suggest that alfaxalone does not suppress the cardiovascular system and therefore would be a safe alternative in aged patients or those with heart-related diseases.

Despite the many benefits of alfaxalone, the volume needed for IM injection is quite large compared to other anesthetics. In this study, for example, the injection volumes for a 10kg animal were 0.5mL of Tel, 7.5mL, 12mL, and 15mL for the Alf7, Alf12, and Alf15 conditions, respectively. This higher volume of injectable drug is comparable in cost to telazol but requires more time for full administration and the likelihood that multiple administration sites be used (i.e., arm, leg, shoulder, etc.). Per recommended guidelines, intramuscular injections should not exceed 0.05mL/kg/site for animals less than 10 kgs, whereas animals greater than 10 kgs may have one injection of 5 mL in one site [19]. This is indeed one of the downsides of alfaxalone use.

Though the volumes for injection were much larger than usually needed for other sedation drugs, we did not observe any localized tissue damage or inflammation associated with injection sites. Similar to Wada, Koyama and Yamashita [18], we did not observe any behavior that would indicate that the larger volume injections were painful (e.g., scratching or grabbing at the injection site, pulling away from injection, etc.). In fact, most animals in the higher dose groups were fully anesthetized (lying prone in cage) before the entire dose could be administered. Future testing should apply subcutaneous route of administration of the larger volumes to compare sedation characteristics and physiological parameters. This alternative administration method has been used previously [20] and would reduce any discomfort related to the larger volumes.

Anecdotally, our technicians noted that under the alfaxalone conditions, muscle spasms were somewhat common during the initial stages of induction and throughout recovery, especially with the lower doses. Though not physically harmful, this effect was unexpected, seemed to occur spontaneously, and resolved itself just as quickly. Future studies to further

characterize this phenomenon are warranted. Additional studies should include aged animals and aim to characterize alfaxalone anesthesia in that specific population.

Overall, results from this study indicate that alfaxalone would be an acceptable alternative anesthetic, compared to ketamine or Telazol, for use during metabolic testing and the most efficient dose that we tested appears to be 12mg/kg administered intramuscularly.

## Supporting information

**S1 Table. Individual subject demographics.**
(DOCX)

## Acknowledgments

The authors would like to thank Dr. Richard Herbert, Ed Tilmont, Danielle Sedlak, and Abbas Hanaee for their technical support.

## Author Contributions

**Conceptualization:** Kelli L. Vaughan, Julie A. Mattison.

**Data curation:** Kelli L. Vaughan, Kielee Toepfer.

**Formal analysis:** Kelli L. Vaughan.

**Methodology:** Kelli L. Vaughan, Julie A. Mattison.

**Project administration:** Julie A. Mattison.

**Resources:** Julie A. Mattison.

**Supervision:** Kelli L. Vaughan, Julie A. Mattison.

**Writing – original draft:** Kelli L. Vaughan, Kielee Toepfer, Julie A. Mattison.

**Writing – review & editing:** Kelli L. Vaughan, Kielee Toepfer, Julie A. Mattison.

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
