## [Decision Letter · Decision Letter 0]

17 May 2024

PONE-D-24-15687Physiological effects of alfaxalone anesthesia on rhesus monkeys during intravenous glucose tolerance testingPLOS ONE

Dear Dr. Vaughan,

Thank you for submitting your manuscript to PLOS ONE. After careful consideration, we feel that it has merit but does not fully meet PLOS ONE’s publication criteria as it currently stands. Therefore, we invite you to submit a revised version of the manuscript that addresses the points raised during the review process.

We look forward to receiving your revised manuscript.

Kind regards,

Jeremy Vance Smedley

Academic Editor

PLOS ONE

Journal Requirements:

"The authors would like to thank Ed Tilmont, Danielle Sedlak, and Abbas Hanaee for their technical support.  This project was supported entirely by the Intramural Research Program, National Institute on Aging, NIH."

Additional Editor Comments:

In this well written manuscript Vaughan et al have evaluated the impacts of alfaxalone/gyclopyrrolate anesthesia on rhesus macaques during intravenous glucose tolerance testing (IVGTT). The study compared the impact on IVGTT results (glucose and insulin), as well as the need for supplemental doses of anesthetic agents (ketamine or alfaxalone), subjective assessment of sedation depth, BP, HR, SPO2, induction time, recovery time between 4 different combination regimens.

All groups included the use of glycopyrrolate (0.015 mg/kg) which has the potential to impact a number of the parameters evaluated and as such should be listed in the title and named prominently in the groups/group designations to avoid confusion for readers. Comparisons were made between Telazol (5mg/kg)/glycopyrrolate (0.015 mg/kg), alfaxalone (7.5 mg/kg)/glycopyrrolate (0.015 mg/kg), alfaxalone (12 mg/kg)/glycopyrrolate (0.015 mg/kg), alfaxalone (15 mg/kg)/glycopyrrolate (0.015 mg/kg). It was assessed the combination of alfaxalone (12 mg/kg) and glycopyrrolate (0.015 mg/kg) was the ideal dose for IVGTT. In addition to the comments by the reviewers it would be important to discuss the potential of glycopyrrolate to impact the parameters as well as details on any issues with repeated large volume IM injections of alfaxalone including cage side or sedated observations of the sites and if any animals have gone to necropsy and had histologic assessment of injection sites. It does appear that others (for example Bertrand et al 2017) have used the subcutaneous route to overcome this potential for discomfort and muscle damage and including additional references and addressing this alternate route would be warranted.

Reviewers' comments:

Reviewer's Responses to Questions

**Comments to the Author**

1. Is the manuscript technically sound, and do the data support the conclusions?

Reviewer #1: Yes

Reviewer #2: Yes

2. Has the statistical analysis been performed appropriately and rigorously? 

Reviewer #1: Yes

Reviewer #2: Yes

3. Have the authors made all data underlying the findings in their manuscript fully available?

Reviewer #1: Yes

Reviewer #2: Yes

4. Is the manuscript presented in an intelligible fashion and written in standard English?

Reviewer #1: Yes

Reviewer #2: Yes

5. Review Comments to the Author

Reviewer #1: This manuscript assessed the effects of alfaxalone anesthesia on rhesus macaques during IV glucose

tolerance testing. The study compared IVGTT results, need for supplemental doses, sedation depth, BP,

HR, SPO2, induction time, recovery time between 4 different combination analgesic regimens. All

included the use of glycopyrrolate (0.015 mg/kg) and either Telazol (5mg/kg), alfaxalone (7.5 mg/kg),

alfaxalone (12 mg/kg), alfaxalone (15 mg/kg). It was assessed the combination of alfaxalone (12 mg/kg)

and glycopyrrolate (0.015 mg/kg) was the ideal dose for IVGTT.

Minor edits/suggestions:

Since Glycopyrrolate was used for each anesthetic study group it should be stated that it is used;

this study should be an assessment of four combination anesthetic regimens throughout this

manuscript (abstract, introduction (Line 81-84), and discussion (Line 249). Considering

glycopyrrolate can cause false increase of HR, increase of temp, hyposalivation, it can greatly

effect the parameters of the results, and the anesthetic groups (Telazol, Alf7.5, Alf12, and Alf15)

should be assessed as a combination anesthetic regimen throughout the study.

Line 19, add word for clarity: “to three different doses of alfaxalone”

Add citations for lines 40, 50, and 71.

Low animal number in study; due to this study evaluating cardiovascular parameters and an

sedative/anesthetic it would be helpful to provide Table 1 with body condition scores of the

group and list out for each monkey its age, sex, weight, and body condition score.

Line 107-108 list route for sedatives

Line 265-267: This is a promising find for alfaxalone as it lowers the anesthesia risk potential and

may reduce the need for extra sedation time to treat effects of salivation

o Cannot really conclude on salivation for alfaxalone-considering all animals were given

Glycopyrrolate which is an antisialogogue, unless an animal group only received

alfaxalone without glycopyrrolate.

Did Alf12 and Alf15 have shorter recovery times because of the actual anesthetic regimen used

or was it more likely due to not receiving supplemental doses? Should discuss this in the

discussion.

Probably should include comparing some of your results with this study in the discussion:

o Wada S, Koyama H, Yamashita K. Sedative and physiological effects of alfaxalone

intramuscular administration in cynomolgus monkeys (Macaca fascicularis). J Vet Med

Sci. 2020 Jul 31;82(7):1021-1029. doi: 10.1292/jvms.20-0043. Epub 2020 May 26. PMID:

32461537; PMCID: PMC7399308.

Reviewer #2: M and M:

o Line 36: I think mentioning that ketamine may cause localized pain and muscle damage is overstated, especially when the animals in this paper are getting up to 15 mL of alfaxalone IM. Additionally, telazol (III) and alfaxalone (IV) are both scheduled drugs as well so I am unsure mentioning that ketamine is controlled is a drawback compared to telazol and alfaxalone, as researchers will also need to obtain DEA licenses/keep up controlled substance logs to utilize these drugs. I suggest considering removing this information on ketamine, especially in the context of this paper.

o Lines 107 and 108: Please provide routes of administered agents. Additionally, though you have provided the brand names/manufacturer in the introduction, I suggest specifying those here as well as the drug concentrations.

o Line 118: Please specify non-invasive vs invasive blood pressure.

o Lines 124: Please specify if the blood was collected through the same catheter that the dextrose was infused through, or via the contralateral saphenous vein?

o Line 163: I suggest specifying if BP was non-invasive or invasive.

o Line 167: In this section, I suggest providing information on significance (ex. “Results were considered statistically significant when the p value was less than 0.05”). Please also indicate information on how data is presented (SEM, SD, etc).

Results:

o Fig 1. I recommend keeping consistency in how you name the groups throughout the manuscript and figures. For example, in line 108 the 7.5 mg/kg group is called “Alf7,” but in Fig 1, the same group appears to be called “A7.5.” Additionally, in the figure description, please list n, specify what data and error bars represent (ex. mean +/- SEM), and provide units for the x-axis.

o Line 196. For “Alf7,” I suggest staying consistent with study group name throughout the rest of the manuscript.

o Fig 2. Please list p values for all significant results in the figure (and/or define asterisks). Additionally, please list n and specify what all error bars signify for all tables.

o Fig 3. Please list p values for all significant results in the figure (and/or define asterisks). Additionally, please list n and specify what all error bars signify for all tables.

Discussion:

o Line 249: I suggest using generic name dexmedetomidine here (rather than dexdormitor).

o Line 289: I realize the purpose of this paper was to simply state the effects of alfaxalone on anesthesia parameters and glucose/insulin levels during the IVGTT. However, I do feel a bit more discussion is warranted in this manuscript on the potential significance of large IM injections on the welfare of the animals, particularly if these injections were repeated over time. Large intramuscular injections are likely painful for the animals and likely induce more local tissue damage than smaller ones (hematomas, necrosis, fibrosis, muscle contracture, nerve damage, etc.). As you would need to give in multiple locations, the animal would need to receive multiple injections in multiple muscle groups and sustain a more prolonged squeeze restraint, which is more stressful than a quick squeeze for a small injection of ketamine or telazol. In light of these welfare concerns, repeated injections of this volume would not be advised long term, and I do not feel the small benefits of alfaxalone in terms of the time to induction, time to first boost, etc outweigh the potential negative effects of the large IM injections that need to be given with alfaxalone. Additionally, some institutional animal care and use committees many not approve the use of such high volumes of IM drugs without good scientific justification.

o Line 290: May I also suggest mentioning cost comparison of drugs at the time of publication, if there is a large difference – for example it appears you will need to use a vial and a half of a 10 mL vial for a standard male rhesus which likely will be much more expensive than using telazol or ketamine + diazepam which are both seemingly acceptable for IVGTT and relatively safe, as you have shown in your previous work.

6. PLOS authors have the option to publish the peer review history of their article (what does this mean?). If published, this will include your full peer review and any attached files.

Reviewer #1: No

Reviewer #2: No

---

## [Author Response · Author response to Decision Letter 0]

11 Jul 2024

Response to Reviewer’s comments:

Editor Comments

1. All groups included the use of glycopyrrolate (0.015 mg/kg) which has the potential to impact a number of the parameters evaluated and as such should be listed in the title and named prominently in the groups/group designations to avoid confusion for readers. 

Because all subjects were pre-treated with glycopyrrolate using a consistent dose, we cannot test the effects of the compound in this setting; it was not designed to do so. We feel that adding it to the title and/or group designations might be misleading because the study was not designed to assess its effects. An additional group that was not given the drug would need to have been included to accurately report the effects of glycopyrrolate. In the current study design, glycopyrrolate is a controlled variable. We have included text to make this point clearer and so there can be no misunderstanding that all subjects were pre-treated with it and the only variable being tested was the anesthetic agent.

2. In addition to the comments by the reviewers it would be important to discuss the potential of glycopyrrolate to impact the parameters as well as details on any issues with repeated large volume IM injections of alfaxalone including cage side or sedated observations of the sites and if any animals have gone to necropsy and had histologic assessment of injection sites. It does appear that others (for example Bertrand et al 2017) have used the subcutaneous route to overcome this potential for discomfort and muscle damage and including additional references and addressing this alternate route would be warranted.

The general effects of glycopyrrolate are listed on page 6. Cage side observations related to animal behavior during injections and injection site observations have been added to the text. The subcutaneous route of administration is now included as an alternative for future studies. No animals were euthanized for this study or near to the time of the injections to allow for histologic assessment. 

Reviewer #1

1. Since Glycopyrrolate was used for each anesthetic study group it should be stated that it is used; this study should be an assessment of four combination anesthetic regimens throughout this manuscript (abstract, introduction (Line 81-84), and discussion (Line 249). Considering glycopyrrolate can cause false increase of HR, increase of temp, hyposalivation, it can greatly effect the parameters of the results, and the anesthetic groups (Telazol, Alf7.5, Alf12, and Alf15) should be assessed as a combination anesthetic regimen throughout the study.

The text was updated to more clearly state that glycopyrrolate was given to all groups (Line 113)

2. Line 19, add word for clarity: “to three different doses of alfaxalone” 

Wording updated for clarity.

3. Add citations for lines 40, 50, and 71. 

References added for lines 40, 50, and 71

4. Low animal number in study; due to this study evaluating cardiovascular parameters and an

sedative/anesthetic it would be helpful to provide Table 1 with body condition scores of the

group and list out for each monkey its age, sex, weight, and body condition score. 

Supplemental Table 1 has been added to the OSF data repository and it details each animal’s body weight, sex, age, and body condition score.

5. Line 107-108 list route for sedatives 

Text updated with route of administration

6. Line 265-267: This is a promising find for alfaxalone as it lowers the anesthesia risk potential and may reduce the need for extra sedation time to treat effects of salivation 

o Cannot really conclude on salivation for alfaxalone-considering all animals were given Glycopyrrolate which is an anti-sialogogue, unless an animal group only received alfaxalone without glycopyrrolate.

Sentence modified to state that alfaxalone may reduce the potential of aspiration during sedation and the need for prophylactic treatment for excessive salivation. All animals were given glycopyrrolate and all but one had notable salivation under telazol anesthesia but not during alfaxalone. Thus, prophylactic treatment may not be needed with alfaxalone. This is a logical interpretation of the data but should be tested in a future study. 

7. Did Alf12 and Alf15 have shorter recovery times because of the actual anesthetic regimen used or was it more likely due to not receiving supplemental doses? Should discuss this in the discussion. 

This is a great point, thank you for raising it. An additional statement has been added to the text to explain that recovery time was not evaluated in relation to the number of supplemental doses required.

8. Probably should include comparing some of your results with this study in the discussion:

Wada S, Koyama H, Yamashita K. Sedative and physiological effects of alfaxalone

intramuscular administration in cynomolgus monkeys (Macaca fascicularis). J Vet Med

Sci. 2020 Jul 31;82(7):1021-1029. doi: 10.1292/jvms.20-0043. Epub 2020 May 26. PMID:

32461537; PMCID: PMC7399308.

Discussion has been added to compare outcomes with the Wada et al. (2020) report. 

Reviewer #2

1. Line 36: I think mentioning that ketamine may cause localized pain and muscle damage is overstated, especially when the animals in this paper are getting up to 15 mL of alfaxalone IM. Additionally, telazol (III) and alfaxalone (IV) are both scheduled drugs as well so I am unsure mentioning that ketamine is controlled is a drawback compared to telazol and alfaxalone, as researchers will also need to obtain DEA licenses/keep up controlled substance logs to utilize these drugs. I suggest considering removing this information on ketamine, especially in the context of this paper. 

Thank you for pointing this out. The portion of the text related to side-effects of Ketamine have been removed.

2. Lines 107 and 108: Please provide routes of administered agents. Additionally, though you have provided the brand names/manufacturer in the introduction, I suggest specifying those here as well as the drug concentrations. 

Text updated with route of administration and the concentration of each drug is now listed.

3. Line 118: Please specify non-invasive vs invasive blood pressure. 

Non-invasive blood pressure is now specified.

4. Lines 124: Please specify if the blood was collected through the same catheter that the dextrose was infused through, or via the contralateral saphenous vein? 

Blood was collected through the same catheter, and this is now specified in the text.

5. Line 163: I suggest specifying if BP was non-invasive or invasive. 

Non-invasive blood pressure is now specified.

6. Line 167: In this section, I suggest providing information on significance (ex. “Results were considered statistically significant when the p value was less than 0.05”). Please also indicate information on how data is presented (SEM, SD, etc).

Thank you for noticing that this was not stated. The text has been updated to reflect the level of statistical significance and SEM reporting.

7. Fig 1. I recommend keeping consistency in how you name the groups throughout the manuscript and figures. For example, in line 108 the 7.5 mg/kg group is called “Alf7,” but in Fig 1, the same group appears to be called “A7.5.” Additionally, in the figure description, please list n, specify what data and error bars represent (ex. mean +/- SEM), and provide units for the x-axis. 

Figure 1 group was updated to Alf7 to match the group designation in the text. Figure legend was also updated to identify that data is area under the curve and error bars are SEM. Units for the x-axis are updated in the figures. 

8. Line 196. For “Alf7,” I suggest staying consistent with study group name throughout the rest of the manuscript. 

Group names in figures have been updated to be consistent with group names in text.

10. Fig 2. Please list p values for all significant results in the figure (and/or define asterisks). Additionally, please list n and specify what all error bars signify for all tables. 

p-values, n, and identification of error bars have been added to the figure.

11. Fig 3. Please list p values for all significant results in the figure (and/or define asterisks). Additionally, please list n and specify what all error bars signify for all tables.

p-values, n, and identification of error bars have been added to the figure.

12. Line 249: I suggest using generic name dexmedetomidine here (rather than dexdormitor). 

Text has been updated to list generic name instead of brand name of drug.

13. Line 289: I realize the purpose of this paper was to simply state the effects of alfaxalone on anesthesia parameters and glucose/insulin levels during the IVGTT. However, I do feel a bit more discussion is warranted in this manuscript on the potential significance of large IM injections on the welfare of the animals, particularly if these injections were repeated over time. Large intramuscular injections are likely painful for the animals and likely induce more local tissue damage than smaller ones (hematomas, necrosis, fibrosis, muscle contracture, nerve damage, etc.). As you would need to give in multiple locations, the animal would need to receive multiple injections in multiple muscle groups and sustain a more prolonged squeeze restraint, which is more stressful than a quick squeeze for a small injection of ketamine or telazol. In light of these welfare concerns, repeated injections of this volume would not be advised long term, and I do not feel the small benefits of alfaxalone in terms of the time to induction, time to first boost, etc outweigh the potential negative effects of the large IM injections that need to be given with alfaxalone. Additionally, some institutional animal care and use committees many not approve the use of such high volumes of IM drugs without good scientific justification. 

We agree, this is a significant point and precisely why studies like ours are so important. Additional discussion points have been added to address potential complications of the large volume injections. Also, an alternative route of administration is discussed and recommended for future study. 

14. Line 290: May I also suggest mentioning cost comparison of drugs at the time of publication, if there is a large difference – for example it appears you will need to use a vial and a half of a 10 mL vial for a standard male rhesus which likely will be much more expensive than using telazol or ketamine + diazepam which are both seemingly acceptable for IVGTT and relatively safe, as you have shown in your previous work.

This is a great point, thank you for raising it. Though the volumes are quite different, per dose, telazol and alfaxalone are fairly inexpensive and comparable in price (both < $50 per dose). A statement to reflect this has been added.

---

## [Editor Report · Decision Letter 1]

16 Jul 2024

Physiological effects of alfaxalone anesthesia on rhesus monkeys during intravenous glucose tolerance testing

PONE-D-24-15687R1

Dear Dr. Vaughan,

We’re pleased to inform you that your manuscript has been judged scientifically suitable for publication and will be formally accepted for publication once it meets all outstanding technical requirements.

Kind regards,

Jeremy Vance Smedley

Academic Editor

PLOS ONE
---

## [Editor Report · Acceptance letter]

17 Jul 2024

PONE-D-24-15687R1 

PLOS ONE

Dear Dr. Vaughan, 

I'm pleased to inform you that your manuscript has been deemed suitable for publication in PLOS ONE. Congratulations! Your manuscript is now being handed over to our production team.

Kind regards, 

on behalf of

Dr. Jeremy Vance Smedley 

Academic Editor

PLOS ONE